# Transcriptome Analysis Reveals the Genes Involved in *Bifidobacterium Longum* FGSZY16M3 Biofilm Formation

**DOI:** 10.3390/microorganisms9020385

**Published:** 2021-02-14

**Authors:** Zongmin Liu, Lingzhi Li, Qianwen Wang, Faizan Ahmed Sadiq, Yuankun Lee, Jianxin Zhao, Hao Zhang, Wei Chen, Haitao Li, Wenwei Lu

**Affiliations:** 1State Key Laboratory of Food Science and Technology, Jiangnan University, Wuxi 214122, China; 7180112023@stu.jiangnan.edu.cn (Z.L.); lilingzhi@sina.cn (L.L.); 6190111076@stu.jiangnan.edu.cn (Q.W.); Faizan@jiangnan.edu.cn (F.A.S.); jxzhao@jiangnan.edu.cn (J.Z.); zhanghao@jiangnan.edu.cn (H.Z.); chenwei66@jiangnan.edu.cn (W.C.); liht@jiangnan.edu.cn (H.L.); 2School of Food Science and Technology, Jiangnan University, Wuxi 214122, China; 3Department of Microbiology & Immunology, Yong Loo Lin School of Medicine, National University of Singapore, Singapore 117545, Singapore; micleeyk@nus.edu.sg; 4International Joint Research Laboratory for Pharmabiotics & Antibiotic Resistance, Jiangnan University, Wuxi 214122, China; 5National Engineering Research Center for Functional Food, Jiangnan University, Wuxi 214122, China

**Keywords:** biofilm, transcriptome, *Bifidobacterium longum*, protein–protein interaction network, SOS response, WGCNA

## Abstract

Biofilm formation has evolved as an adaptive strategy for bacteria to cope with harsh environmental conditions. Currently, little is known about the molecular mechanisms of biofilm formation in bifidobacteria. A time series transcriptome sequencing analysis of both biofilm and planktonic cells of *Bifidobacterium longum* FGSZY16M3 was performed to identify candidate genes involved in biofilm formation. Protein–protein interaction network analysis of 1296 differentially expressed genes during biofilm formation yielded 15 clusters of highly interconnected nodes, indicating that genes related to the SOS response (*dnaK*, *groS, guaB, ruvA, recA*, *radA*, *recN*, *recF*, *pstA*, and *sufD*) associated with the early stage of biofilm formation. Genes involved in extracellular polymeric substances were upregulated (*epsH*, *epsK, efp*, *frr*, *pheT*, *rfbA*, *rfbJ*, *rfbP*, *rpmF*, *secY* and *yidC*) in the stage of biofilm maturation. To further investigate the genes related to biofilm formation, weighted gene co-expression network analysis (WGCNA) was performed with 2032 transcript genes, leading to the identification of nine WGCNA modules and 133 genes associated with response to stress, regulation of gene expression, quorum sensing, and two-component system. These results indicate that biofilm formation in *B. longum* is a multifactorial process, involving stress response, structural development, and regulatory processes.

## 1. Introduction

Biofilm formation represents a unique protective mode of bacterial growth in which bacterial cells are structurally organized and their tolerance to several hostile conditions is dramatically improved [1]. The beneficial effects of probiotic strains in the form of biofilms have been verified, including their increased resistance to gastric pH, temperature, and mechanical forces compared to their planktonic counterparts [2]. In addition, biofilms formed by probiotic biofilms can be potentially used to control the growth of spoilage and pathogenic bacteria in industrial and medical setups [3]. *Bifidobacterium* species are commonly used as probiotics for human consumption, given their beneficial relationship in human gastrointestinal health and nutrition [4,5]. The ability of *Bifidobacterium* to form biofilms on abiotic surfaces (stainless steel, glass, polystyrene, or complex food matrices) has been studied, and the results suggests that biofilm formation is driven by the properties of both the selected strains and the carriers [6].

Biofilm formation is often triggered in response to environmental stresses, such as nutrient starvation, oxidative stress, pH and bile [7]. Oxidative stress is generated by both metabolic processes and diverse environmental stress factors, known to generate reactive oxygen species (ROS), which are triggers of the SOS response [8]. The SOS response is a global response to DNA damage, and two proteins play key roles in the regulation of this response: LexA and RecA [9]. Oxidative stress caused by oxygen treatment (3%, *v*/*v*) contributes to *B. longum* BBMN68 biofilm formation [10]. The process of biofilm formation can be described in the following stages: reversible to irreversible attachment, development of microcolonies, maturation of biofilm architecture, and biofilm dispersion [11]. Type IV tad pili and quorum sensing (QS) are important for the early stages of biofilm formation [12,13]. The functions of tad IV pili include adhesion, biofilm formation, motility, and molecule exchange [14]. Type IV pili are important surface appendages that are central to the surface-sensing mechanism in the early stages of biofilm formation [13]. QS, two-component systems and nucleotide second messengers c-di-GMP [15,16] are considered as the main regulators controlling extracellular polymeric substances (EPS) production during biofilm formation [17]. Two-component systems consist of a histidine kinase and a cognate response regulator [18]. The two-component signal transduction system plays an important role in bacteria to monitor internal or environmental signals and then translate these stimuli into appropriate cellular responses, which is also involved in bacterial biofilm formation [19]. The EPS can lead to the development of biofilm microcolonies by promoting cell adhesion to solid substrates and cohesion among bacterial cells [20]. To date, three types of AI-2 receptor have been identified: LuxP, LsrB, and RbsB in *Vibrio harveyi*, *Salmonella enterica* serovar *Typhimurium*, and *Escherichia coli*, respectively [21]. All *Bifidobacterium* strains sequenced harbor luxS genes, which are involved in the production of the classic QS signaling molecule autoinducer-2 (AI-2) [22,23,24], yet no AI-2 receptor has been reported in this genus [25]. Despite a vast amount of research on biofilm formation [26], its mechanism has not yet been fully elucidated in many bacterial species, and it is especially true for *Bifidobacterium* species [27].

In this study, we established a bifidobacterial biofilm fermentation system and investigated transcriptional changes in *B. longum* FGSZY16M3 in the biofilm and planktonic states to reveal candidate genes involved in biofilm formation. The potential protein–protein interaction (PPI) network was used to identify potential interactions among differentially expressed genes (DEGs) during biofilm formation. Using the key genes selected from the PPI network as references, genes involved in biofilm formation were further mined through weighted gene co-expression network analysis (WGCNA). Our analysis provides insights into the mechanism of biofilm formation in bifidobacteria.

## 2. Materials and Methods

### 2.1. Planktonic and Biofilm Culture

*B. longum* FGSZY16M3 was isolated from human feces and stored in the state key laboratory of food science and technology (Jiangnan University, Wuxi, China). The strain was cultured anaerobically at 37 °C in MRS broth supplemented with 0.5 g/L L-cysteine hydrochloride-monohydrate. Overnight *B. longum* FGSZY16M3 culture was inoculated (4%, *v*/*v*) into Erlenmeyer flask containing 60 mL MRS medium, and the culture was grown at 37 °C with 5% (*v*/*v*) oxygen [28] and 120 rpm for 86 h. The hollow glass microspheres (GM) with an average diameter of 50 μm and density of 0.48 g/cm^3^ (Sinosteel maanshan general institute of mining research Co., LTD, Maanshan, China) were added to culture media with 1 g/L for biofilm culture [29,30]. No GM added groups were used as planktonic culture. To obtain biofilm cells, we used 40 μm cell strainer (NO. F613461, Sangon Biotech Co., Ltd., Shanghai, China) to filter the GM group culture, and then washed the particles to wash away planktonic cells. The colony-forming units (CFU) and pH values in the GM and control culture group were determined at 16, 34, 68, and 86 h during fermentation [31]. For GM culture group CFU counting, samples were vortexed for 30 s, sonicated for 10 s (Vibra Cell Model VCX150PB, Sonics & Materials Inc., Danbury, CT, USA) and vortexed again for 30 s in order to disperse biofilm cells into the suspension [32]. To avoid contamination during sampling, we made a lot of parallels with Erlenmeyer flasks at the beginning. At each time point, two fresh Erlenmeyer flasks were taken for sampling as repeats.

### 2.2. Transcriptomic Analysis

Illumina sequencing of the pooled RNA-seq libraries yielded 14 FASTQ files of sequences (16 hC, 16 hG, 34 hC, 34 hG, 68 hC, 68 hG, 86 hG; C: the control culture group without GM; G: the culture group in the presence of GM; two biological replicates per condition). Transcriptome data of *B. longum* FGSZY16M3 during biofilm formation have been deposited in the National Center for Biotechnology Information Search database (NCBI) under BioProject accession code PRJNA680454. We preprocessed paired-end reads using fastp [33], which can perform adapter trimming, quality filtering and quality control. Reads were aligned to *B. longum* FGSZY16M3 genomes using HISAT2 v2.20 [34]. The output SAM files were converted to BAM using SAMtools v1.10 (England, UK) [35]. Raw read counts were created using featureCounts (R package) [36].

### 2.3. Functional and Pathway Analysis of Differentially Expressed Genes (DEGs)

Differential expression analysis of biofilm and planktonic cells was performed using the DESeq2 [37]. Specified pairwise transcriptome comparisons were performed to identify the main differentially expressed genes (DEGs) with an absolute value of log2 (fold change) >1 and an adjusted *p*-value threshold of <0.5. Venn diagram was used to show the comparison and overlap between DEGs in different biofilm formation stages [38]. Gene Ontology (GO) and Kyoto Encyclopedia of Genes and Genomes (KEGG) pathway analyses were conducted using clusterprofiler (R package) [39] for DEGs obtained from different stages.

### 2.4. Protein–Protein Interaction (PPI) Network Analysis

The search tool for retrieval of interacting genes (STRING) (https://string-db.org, accessed on 12 January 2021) database, which integrates both known and predicted PPI networks, can be applied to predict functional interactions of proteins [40]. To seek potential interactions between DEGs in biofilm formation stages, the STRING tool was employed [41]. Active interaction sources, including text mining, experiments, databases, and co-expression as well as species limited to “*Bifidobacterium longum*” and an interaction score >0.4 were applied to construct the PPI network. Cytoscape software version 3.8.0 was used to visualize the PPI network (San Diego, CA, USA) [42]. To detect highly connected regions of the network, MCODE software was used. In the network, the nodes correspond to the proteins and the edges represent the interactions. Among the non-redundant 1296 DEGs, only 635 DEGs had annotated gene names. Uploading 635 DEGs to the STRING online database, an interaction network was generated with 228 nodes and 3344 edges.

### 2.5. Identification of Weighted Gene Co-Expression Network Analysis (WGCNA) Modules

The highly co-expressed gene modules were inferred using WGCNA [43], an R package. WGCNA network construction and module detection was conducted using an unsigned type of topological overlap matrix (TOM), a power β of 4, a minimal module size of 30, and a branch merge cut height of 0.25. Gene network files selected according to the suitable WGCNA edge weight values were used as the Cytoscape input file (‘Yellow’, ‘Pink’ and ‘Turquoise’ with WGCNA edge weight over 0.81, 0.70 and 0.77, respectively). To identify the key hub genes within each module, we visualized the gene network using Cytoscape software. The node circle size is positively correlated with the number of genes that are partnered within interactions. The hub genes refer to the genes with the biggest node size.

### 2.6. Identification of Genes Involved in AI-2, Tad Pili and Signal Peptides

To identify an AI-2 binding protein in *Bifidobacterium*, we download LsrABCD, RbsABCD and luxPQ protein sequences from NCBI (https://www.ncbi.nlm.nih.gov/protein/, accessed on 12 January 2021) as query sequences to perform a blastp (protein-protein BLAST) analysis (cut-off value 1e-30) [44,45,46]. Signal peptides were identified by SignalP 5.0, which can discriminate three types of signal peptide (http://www.cbs.dtu.dk/services/SignalP/, accessed on 12 January 2021): Sec/SPI, Sec/SPII and Tat/SPI.

### 2.7. Statistical Analysis

Data were analyzed using the RStudio (v3.5.0) environment (https://www.r-project.org/index.html, accessed on 12 January 2021). The R package ComplexHeatmap (v2.5.1) (https://jokergoo.github.io/ComplexHeatmap-reference/book/, accessed on 12 January 2021) [47] was used to process the heat-map. The package ggplot2 (v3.3.2) (https://ggplot2.tidyverse.org/reference/, accessed on 12 January 2021) was used for graphical representation of data. The difference was calculated using *t*-test and considered statistically significant at *p* < 0.05.

## 3. Results

### 3.1. Bifidobacterium longum (B. longum) FGSZY16M3 form Biofilm on Glass Microspheres (GM)

In this study, we used hollow GM (with an average diameter of 50 μm and density of 0.48 g/cm^3^) as carriers in the fermentation system, which can intuitively reflect the biofilm formation process. Figure 1 indicates that *B. longum* FGSZY16M3 can form biofilm on the surfaces of GM. Under the culture condition of 120 rpm, cells did not aggregate together in the control group (Figure 1a,c,e,g), which we collected as planktonic cells. Since the hollow GM were small, they were dispersed into the culture medium when shaken gently, resulting in an increase in turbidity compared with control group (Figure 1a,b). However, after 16 h dynamic culture, a great number of white particles floated in the GM culture group when shaking stopped (Figure 1b), indicating the cells were attached to the GM surface. The volume of the particles gradually increased during 16–34 h (Figure 1d,f). Some particles began to disperse into smaller ones in GM culture after 68 h (Figure 1h). The pH of the GM group was lower than that of the control group (Figure 1i). The added GM provided an adsorption medium for the cells and promoted their proliferation, resulting in more acid production than the control group. In return, a certain number of cells and low environmental pH values may contribute to biofilm formation. There were 6.25 × 10^2^ and 1.25 × 10^7^ CFU/mL in the control culture and the GM culture at 86 h (Figure 1j), respectively (*p* < 0.05), indicating that the cells were more resistant to stress after biofilm formation.

### 3.2. DEGs during Biofilm Formation

To describe changes in the biofilm process, we divided biofilm formation process into three stages: the early stage of biofilm formation or S1, maturation of biofilm architecture or S2, and biofilm dispersion or S3. Transcriptome analysis was performed on biofilm samples collected from these three stages and specified pairwise transcriptome comparisons (S1: 16 hG vs. 16 hC; S2: 34 hG vs. 34 hC; S3: 68 hG vs. 68 hC) were performed to identify the main DEGs during biofilm formation. This analysis yielded 1842 DEGs, including 2 DEGs in S1 (2 upregulated) (Figure 2a), 992 DEGs in S2 (477 upregulated, 515 downregulated) (Figure 2b), and 848 genes in S3 (484 upregulated, 364 downregulated) (Figure 2c). The Venn diagram shows the 1296 non-redundant DEGs during the biofilm formation (Figure 2d). A total of 63.8% genes (1296/2032) were differentially expressed during biofilm formation. There were only two upregulated DEGs (*draG* and *amyE*) in S1 (Figure 2a). Blo01|peg.184 (*amyE*, bacterial extracellular solute-binding protein) was predicted to function as a LIPO(Sec/SPII)-type signal peptide. Remarkably, S2 involved the most DEGs, of which 544 DEGs were shared with S3, which indicated that S2 may be a critical stage for biofilm formation.

### 3.3. Network Analysis of the DEGs and 15 Clusters were Identified

To investigate the function of DEGs during biofilm formation, we used the STRING database to identify potential interactions between them. A PPI network between DEGs was constructed and 15 clusters of highly interconnected nodes were identified by MCODE (Figure 3). Nine of the 15 clusters were related to substance synthesis: cluster 1 was related to translation and peptide biosynthetic processes, clusters 2 and 3 were related to aminoacyl-tRNA biosynthesis for protein translation, cluster 5 was related to peptidoglycan biosynthesis, clusters 8, 12, and 13 were related to biosynthesis of amino acids metabolites, and clusters 9 and 10 were related to ribonucleotide biosynthesis. Five clusters, namely clusters 4, 6, 7, 14, and 15, were related to the stress response.

Remarkably, there were five clusters (4, 6, 7, 14, and 15) mainly related to the SOS response (Table 1), indicating the ability of *B. longum* to respond to several environmental stresses. *B. longum* FGSZY16M3 possesses LexA-RecA repressor-activator proteins of the SOS response system. Among these SOS response genes, *guaB* (the main gene in cluster 4), *recF*, *groS*, *sufD*, *atpD*, *atpF*, and *atpH* were upregulated in S2, while *recA*, *dnaK* (response to oxidative stress), and *recN* (involved in recombinational repair of damaged DNA) were upregulated in S3. The gene *clpB* was upregulated in S3, and was associated with *grpE* (Figure 3). There were no DEGs related to stress response in the S1 stage, while several key DEGs related to SOS response were upregulated in the S2 and S3 stages, indicating the SOS response was involved in biofilm formation.

Approximately 25% of DEGs were grouped in cluster 1, and the most significant biological processes and pathways were associated with translation and peptide biosynthetic processes. Most of these genes were upregulated in the S2 stage, indicating a stimulation in EPS production following an increase in translation speed, which may contribute to biofilm formation. In this cluster, we identified several key genes, including *pheT*, *secY*, *frr*, *rpmF*, *efp*. Genes (*argS*, *alaS*, *hisS*, *ileS*, *leuS*, *pheS*, *proS*, *trpS*, and *valS*) in two clusters (clusters 2 and 3) were related to aminoacyl tRNA synthetases, which play a role in protein biosynthesis. The hub gene *guaA* in cluster 2 is involved in many cellular processes, including nucleobase-containing compound metabolic process, glycosyl compound metabolic process, ribonucleoside monophosphate biosynthetic process, and organonitrogen compound biosynthetic process. The hub gene *polA* in cluster 3 exhibits polymerase activity and 5’-3’ exonuclease activity, which associated with base excision repair, nucleotide excision repair, and DNA replication. There were three clusters involved in amino acids metabolites included cluster 8 (valine, leucine, and isoleucine biosynthesis: *ilvD*, *leuB*, *leuC*, and *leuD*), cluster 12 (lysine biosynthesis: *askB*, *dapA*, and *dapB*), and cluster 13 (lysine biosynthesis: *argD*, *dapD*, and *lysA*). Notably, *yajC* (preprotein translocase subunit) in cluster 8 involved in protein export and QS was downregulated in S3.

Genes in two clusters (clusters 9 and 10) were associated with ribonucleotide biosynthetic processes (*dnaA*, *pgk*, *pyrG*, *carA*, *purT*, and *pyrF*). DnaA binds to ATP and acidic phospholipids, in a two-component system. *pyrG* regulates intracellular CTP levels through interactions with four ribonucleotide triphosphates, *pgk* (a phosphoglycerate kinase), and *topA* (a negative regulator of RNA and nitrogen compound metabolic process) were upregulated in S2, while *carA* (belonging to the CarA family), *pyrF* (essential for recycling GMP and indirectly, cGMP), and *purT* (transferase activity, transferring phosphorus-containing groups) were upregulated in S3. Peptidoglycan biosynthesis (*murA*, *murB, murD*, *and murE*) and biosynthesis of amino acids (*hisD, hisE, hisF, hisG*, and *hisI*) were identified in cluster 5. Interestingly, in addition to participating in amino acid synthesis, *hisF* (phosphoribosyl-ATP pyrophosphohydrolase) can respond to external stimuli and is involved in QS.

A schematic representation of the tad locus from *Bifidobacterium* is shown in Figure 4. The tad IV pili genes *tadA*, *tadB*, *tadC*, *tadF, and tadZ* were upregulated in S3. The gene *tadA*, which is associated with *trmD* (a gene that belongs to the RNA methyltransferase TrmD family involved in gene expression) in cluster 1, was upregulated in S3 (Figure 3). Similarly, *guaA* in cluster 2 was upregulated in S2, and *polA* (downregulated in S3) was upregulated in S3. Through the functional relationship and expression changes of these genes, it can be inferred that *trmD*, *guaA*, and *polA* genes may participate in the formation and dispersion of biofilm by regulating tad IV pili genes, which control bacterial movement.

### 3.4. Building and Detection of Functional Modules

To further investigate the genes that are related to biofilm formation, a scale-free gene co-expression network was performed with 2032 transcript genes, leading to the identification of nine WGCNA modules (Figure 5a). Changes in pH and cell numbers reflected the process of biofilm formation (Figure 1). Analysis of the module-trait relationships (Figure 5b) revealed that cell numbers were highly correlated with the yellow module of 263 genes (*r* = 0.83, *p* < 0.01) and pink module of 94 genes (*r* = 0.72, *p* < 0.01). The turquoise module of 425 genes appeared to be positively associated with pH (*r* = 0.65, *p* < 0.05). *dnaB, nudG, recF*, and *relA* in the yellow module, *mazG*, *pafA*, *sdhA*, and *nfo* in the pink module, and *typA* and *ung* in the turquoise module were related to stress. *recF* and *relA* in the yellow module and *aroP* in the pink module were related to cell communication. *dppD, fadD, livF, livG, livK*, Blo01|peg.1382, and Blo01|peg.1383 in the yellow module, *luxE*, *sapF*, *tsaD*, and Blo01|peg.1384 in the pink module, *aroG* in the turquoise module, and Blo01|peg.1342 in the grey module were related to QS.

In the yellow module (Figure 5c), we identified several hub genes, including *dut* (involved in nucleotide metabolism), *ilvB* (thiamine pyrophosphate enzyme), *recF* (involved in DNA replication and normal SOS inducibility), *fadD* (an AMP-binding enzyme), and *rpsE* (ribosome). The key hub genes detected in the turquoise module (Figure 5d) were Blo01|peg.81 (glycosyl transferase, family 2), Blo01|peg.1740 (a SP(Sec/SPI) signal peptide), Blo01|peg.1968 (Protein of unknown function (DUF4012), microbial metabolism in diverse environments), and *ywrO* (flavodoxin-like fold, biosynthesis of secondary metabolites). In the pink module (Figure 5e), we identified several hub genes, including *lacL* (galactose metabolism), *ppk* (oxidative phosphorylation), *araD* (microbial metabolism in diverse environments), *sapF* (ATPases associated with a variety of cellular activities, quorum sensing), and *hisD* (oxidoreductase activity, acting on CH-OH group of donors). These results indicated that the coexpressed gene modules have different functions during biofilm formation.

### 3.5. Genes Involved in Biofilm Formation

After PPI network analysis, genes in terms of SOS response and EPS production, response to stress, regulation of gene expression, QS, two-component system may be involved in biofilm formation process. Genes with similar expression patterns may have similar functions. It is possible to predict whether a new gene is also involved in the biofilm process using the known genes involved in biofilm formation. The fragments per kilobase of exon per million mapped fragments (FPKM) was used to conduct the hierarchical clustering analysis of transcript abundance, and 2032 genes were divided into nine clusters based on the WGCNA modules (Figure 6). The upregulated DEGs in the S2 stage were mostly in yellow, green, black, and red modules, and the upregulated DEGs in S3 stage were brown and blue, whereas the downregulated DEGs in S2 and S3 were related to the turquoise module. The genes in the pink and gray modules were not differentially expressed genes, and the expression levels were low in each sample. To further identify genes involved in biofilm formation, relevant WGCNA modules were selected for functional analysis. There were 6, 18, 12, 15, 26, 19, and 37 genes associated with response to stress, regulation of gene expression, QS, and two-component system identified in black, blue, brown, green, red, yellow, and turquoise, respectively.

EPS immobilize cells and keep them in close vicinity, leading to biofilm formation [48]. EPS are mainly comprised of polysaccharides, proteins, nucleic acids, and lipids, and their production usually promotes biofilm formation. In the S2 stage, following genes related to exopolysaccharide biosynthesis were upregulated (3.22, 1.40, 2.32, 3.12 and 2.46-fold, respectively): *rfbA*, *rfbJ*, *rfbP*, *epsK* and *epsH*. In addition, we sought to seek genes involved in biofilm regulatory processes. AI-2 is unique to QS signaling molecules, as it is produced and recognized by a wide variety of bacteria and thus facilitates interspecies communication [7]. In the black module, we found that *luxS* involved in the synthesis of AI-2 was highly expressed (120.65) in 34 hG sample, and RbsB-type receptor homology genes were identified (identity >30%, and *E* value <10^−30^) in *B. longum* FGSZY16M3 (Table 2) and were also highly expressed in the early stages of biofilm formation (Table 3). Two-component systems generally function as global regulators of gene expression in response to environmental conditions [49]. The gene *senX3* related to two-component systems was found in the turquoise module. A genomic analysis of *B. longum* FGSZY16M3 showed that it harbored a complete SenX3-RegX3 two-component system. Blo01|peg.664 (*senX3*, A0QR01.1, 40.58%, 6.12 × 10^−63^) encoding a sensor histidine kinase and Blo01|peg.665 (*regX3*, Q9F868.1, 60.87%, 7.61 × 10^−100^) encoding a response regulator. Five genes related to stress response and gene expression regulation, including *dnaK*, *groS*, *rplM*, *whiB*, and *zur*, gradually increased in the early and mature stages of the biofilm, and decreased during the biofilm dispersion (Table 3). These results indicate that biofilm formation of *B*. *longum* is a multifactorial process involving specific structural genes and regulatory processes, representing a resistance strategy in harsh living conditions.

## 4. Discussion

We successfully collected samples from *B. longum* FGSZY16M3 under biofilm and planktonic conditions during fermentation and specified time-course and pairwise transcriptome comparisons were made to represent the main DEGs at each stage that included: S1 (16 hG vs. 16 hC), the early stage of biofilm formation; S2 (34 hG vs. 34 hC), the stage of biofilm maturation; and S3 (68 hG vs. 68 hC), the stage of biofilm dispersion. The PPI network analysis and WGCNA indicated that genes associated with SOS response, EPS production, response to stress, regulation of gene expression, QS, and two-component system may be involved in biofilm formation.

The STRING database contains information from various sources, including experimental data, computational prediction methods, and public text collections, and is updated on a regular basis [50]. This database was used to analyze DEGs at different stages of biofilm formation, which remained helpful in identifying key genes and building gene interaction networks. Thus, potential interactions among DEGs were constructed using the STRING tool, and 15 clusters of highly interconnected nodes were identified to be associated with SOS responses and EPS production (Figure 3). However, there are some factors that may have affected the results. For instance, the transcriptome analysis of biofilms formed at three different time points may have missed some important genes related to biofilm formation that are not significant, there may be some errors in annotation or some genes may have no annotated names, and some genes are strain-specific [41]. Despite 1296 non-redundant genes that were differentially expressed, the final PPI network only had 228 genes, which means that the role of most genes in biofilm formation is still unknown. Using the key genes selected from the PPI network as references, the biofilm related genes in the strains were further mined through WGCNA [51]. WGCNA was performed to decompose 2032 coding genes into nine functional modules (Figure 5a). Cell numbers were highly positively correlated with yellow and pink modules, and *recF*, *relA*, and *aroP* were related to cell communication, which may also be associated with QS. The turquoise module was positively associated with pH, indicating that the two genes *typA* and *ung* were regulated in response to pH stress (Figure 5d). *typA* has been previously reported to be involved and required for the survival of *Sinorhizobium meliloti* under certain stress conditions, including pH stress [52]. We combined the DEG-based PPI network and co-expression analysis based on 2032 genes (Figure 6). It was found that the SOS response, tad IV pili, EPS production system, AI-2/quorum sensing, and senX3-regX3/two-component system may be related to biofilm formation in *B. longum*.

A previous study suggested that bacterial interactions with abiotic surfaces can lead to SOS induction [53]. Thus, attaching to the glass microsphere surface may trigger the SOS response. The mediated flow speed can affect biofilm growth, and hydrodynamics affect microbial density [54]. *B*. *longum* FGSZY16M3 was cultured at 120 rpm, which may have caused hydrodynamic stress to the growing cells. In addition, bifidobacteria are also susceptible to oxygen stress when cultured in vitro. Oxygen-induced DNA damage leads to activation of LexA-RecA in *B*. *longum* BBMN68, which subsequently protects it from oxidative stress [10]. *B. longum* FGSZY16M3 possesses LexA-RecA repressor-activator proteins, involved in the regulation of the SOS system [10]. The biofilm life cycle, including initial adhesion, biofilm maturation, cell death, and dispersal, is affected by SOS [55].

Type IV tad pili and QS are important for the early stages of biofilm formation [12,13]. The product of *fadD* is a long-chain fatty acyl-CoA ligase, which reportedly plays an important role in surface motility and biofilm development in *Sinorhizobium meliloti* [56], which indicates that FadD-related compounds may have a possible role in surface motility and biofilm formation in bacteria. Exposure to exogenous fatty acids has been reported to affect growth, membrane permeability, and biofilm formation in *Klebsiella pneumoniae* [57]. From the changes in the expression of the genes of *fadD* (Bl01|peg.237) in different stages, it may be related to tad IV pili movement and biofilm formation in *B. longum* FGSZY16M3. Tad IV pili are important surface appendages that are central to the surface-sensing mechanism in the early stages of biofilm formation [13]. When the bacterial cell number reaches a certain concentration threshold, genes related to quorum sensing begin to express, including the genes related to AI-2 (*luxS*) and some autoinducer peptide (AIP) signaling molecules, such as *amyE* (Bl01|peg.1841) and *livK* (Bl01|peg.329). AIP signals and AI-2 are used by Gram-positive bacteria [22]. AI-2 affects biofilm formation in *B. longum* NCC2705, and all *Bifidobacterium* strains sequenced harbor the *luxS* gene (which involved in AI-2 production) [24]. Homology genes for RbsB-type receptors were identified in *B. longum* FGSZY16M3 and they were expressed, indicating that AI-2 serves may as a QS signal molecule (Table 2 and Table 3). The precursor peptide AIs involved in QS are modified and transported out of the cell by either the twin arginine translocation (Tat) pathway or the ABC transport system [58]. When the concentration of the peptide AIs reaches the threshold value, the sensor kinase protein is activated and phosphorylates the response regulator. Transport systems often recognize diverse substrates, including materials required for biofilm formation. For instance, the ABC transport system dppBCDF is responsible for the uptake of dipeptides and tripeptides [59].

Genes that produce EPS in *B. longum* FGSZY16M3 (*rfbA*, *rfbJ*, *rfbP*, *epsK*, and *epsH*) were vital for microcolony formation. Increased EPS production also reflects a highly adaptive response of biofilms to environmental stress factors such as high shear stress [60]. Moreover, EPS production is often involved in the oxidative stress response [61]. Almost all gram-positive organisms possess two or more *yidC* genes [33]. Three *yidC* genes (Bl01|peg.766, Bl01|peg.767, and Bl01|peg.1516) were identified in *B. longum* FGSZY16M3. There is growing evidence that *yidC* may play a role in biofilm formation. YidC proteins in bacteria function as membrane integral chaperone/insertases associated with the SecYEG translocon. Elimination of *yidC* paralogs in *Streptococcus* causes changes in the cell envelope and glucan production, which ultimately disrupts EPS composition and biofilm development [62]. In addition, YidC was identified as a target of the compound capable of inhibiting biofilm formation in *Staphylococcus aureus* [63]. The expression of *yidC* was significantly influenced by pH and starvation in *Vibrio alginolyticus*, and bacterial adhesion was significantly decreased after *yidC* gene silencing [64]. The gene *relA* reportedly plays an important role in biofilm formation and acid tolerance in *Streptococcus* [65]. Upregulation of the two genes *yidC* and *relA* in *B. longum* FGSZY16M3 cells indicated nutritional stress and low pH conditions.

## 5. Conclusions

In conclusion, a time series global transcription profiling of *B. longum* FGSZY16M3 showed transcriptional changes when cultivated under biofilm and planktonic conditions. Using DEG-based PPI network analysis and WGCNA, the present study suggested that genes related to SOS response and EPS production, AI-2/QS and senX3-regX3/two-component may be involved in biofilm formation. These findings provide a theoretical framework for further research on the mechanism of biofilm development in bifidobacteria. However, biofilm formation is a complex process, involving stress response, structural development, and regulatory processes. Integrated transcriptomic, metabolomics and proteomic analysis of bifidobacteria under different stress response will be needed to verify the roles of identified genes during biofilm formation.

## Figures and Tables

**Figure 1 microorganisms-09-00385-f001:**
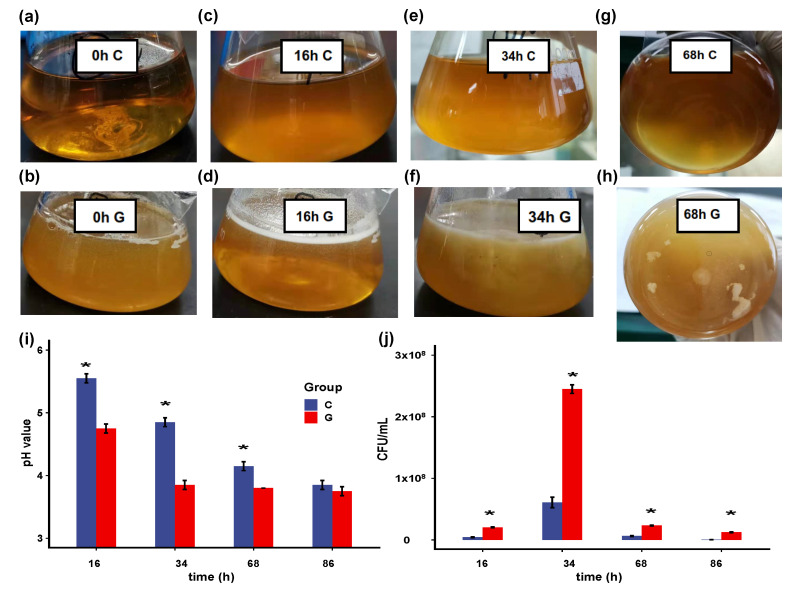
*B. longum* FGSZY16M3 form biofilm on glass microspheres (GM). Control group at 0 h (**a**); 16 h (**c**); 34 h (**e**); and 68 h (**g**); GM group at 0 h (**b**); 16 h (**d**); 34 h (**f**) and 68 h (**h**); (**i**) pH values of control culture and GM culture. Significance is expressed in comparison with the controls at the same time (* *p* < 0.05 are significantly different); (**j**) Cell number of control culture and GM culture.

**Figure 2 microorganisms-09-00385-f002:**
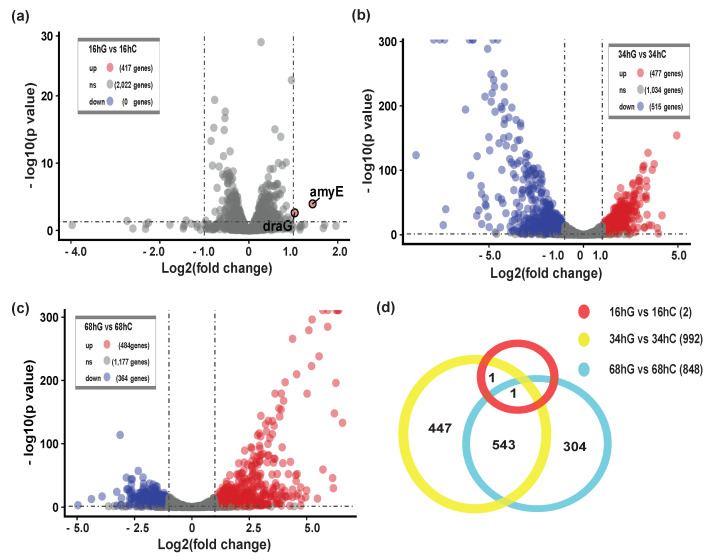
Differentially expressed genes (DEGs) during biofilm formation. (**a**) The volcano plot of S1 showing the upregulated (red points) and downregulated (green points) DEGs between 16 hG and 16 hC. G, the culture group in the presence of GM. C, the control group without GM; (**b**) The volcano plot of S2 showing the DEGs between 34 hG and 34 hC; (**c**) The volcano plot of S3 showing the DEGs between 68 hG and 68 hC; (**d**) Venn diagram representation of the number of DEGs during biofilm formation.

**Figure 3 microorganisms-09-00385-f003:**
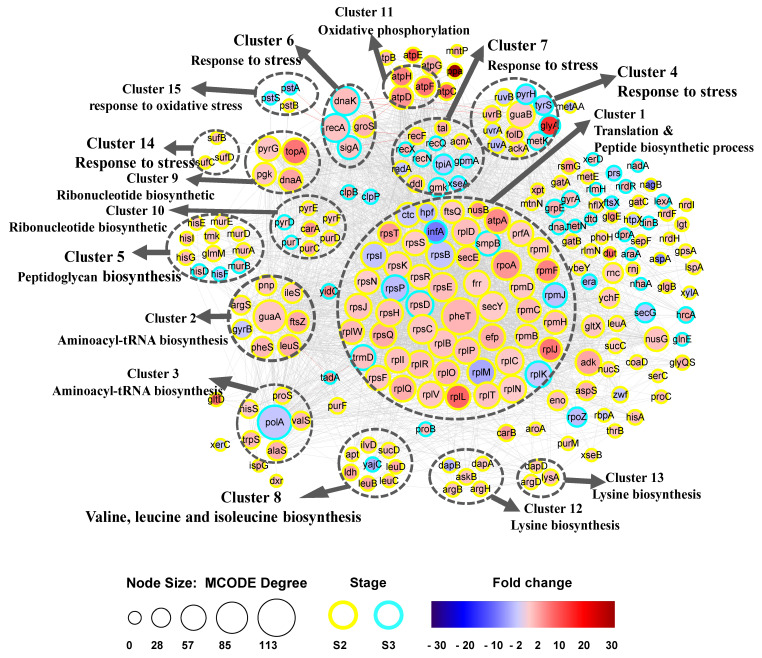
Biofilm formation protein–protein interaction (PPI) network. 15 distinct functional clusters were detected in DEGs during biofilm formation. Each cluster is a set of highly-connected nodes and is illustrated in a circle. The size of the node was determined by MCODE degree. Yellow was S2, and turquoise was S3. Fold change was represented with red and blue color shade. Red was upregulated, and blue was downregulated.

**Figure 4 microorganisms-09-00385-f004:**
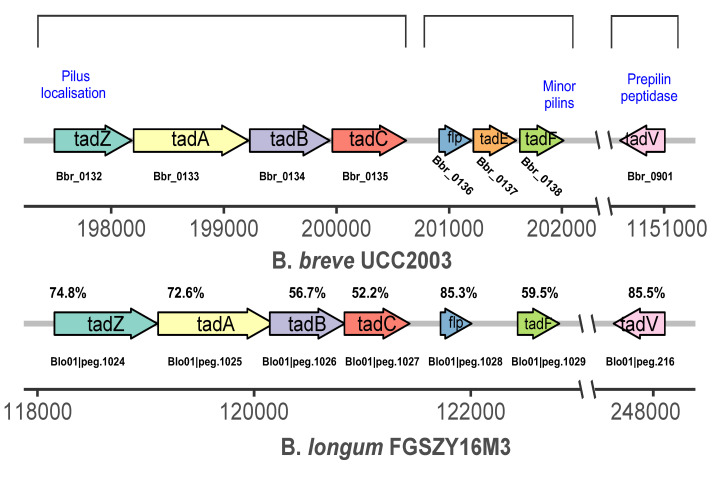
Schematic representation of the tad locus from *B. longum* FGSZY16M3. Each arrow represents an open reading frame with the gene number given above the arrow and the gene name given within the arrow. The functions of the encoded proteins are indicated below the arrow. The levels of amino acid identity compared with *B. breve* UCC2003 (expressed as percentages) are indicated.

**Figure 5 microorganisms-09-00385-f005:**
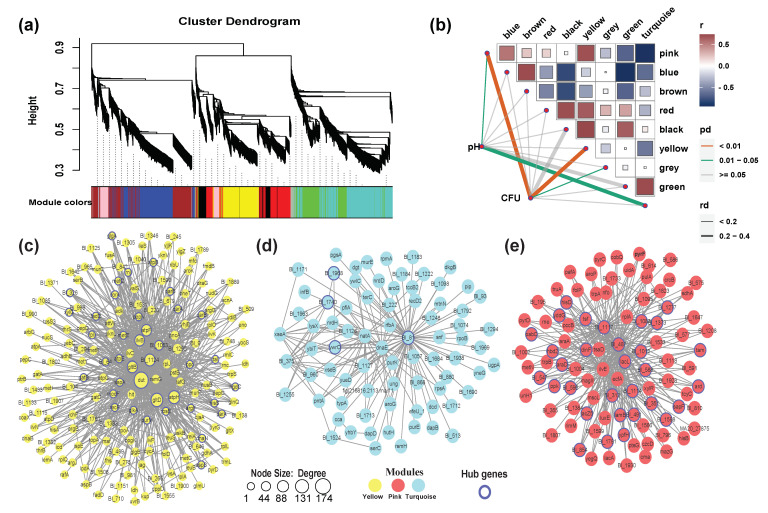
Weighted gene co-expression network analysis (WGCNA) of expressed genes. (**a**) Hierarchical cluster tree showing nine modules of co-expressed genes. Each of the 2032 genes is represented by a leaf in the tree, and each of the nine modules by a major tree branch. The lower panel shows modules in designated colors, such as ‘Blue’, ‘Pink’, ‘Turquoise’ and others. Note that module ‘Grey’ is for unassigned genes; (**b**) Module-colony-forming units (CFU)/pH correlations and corresponding P-values. Co-expression network analysis of yellow (**c**); turquoise (**d**); and pink (**e**) modules. The size of the node circle is positively correlated with the number of interacting gene partners. The genes marked in blue present the hub genes of each module.

**Figure 6 microorganisms-09-00385-f006:**
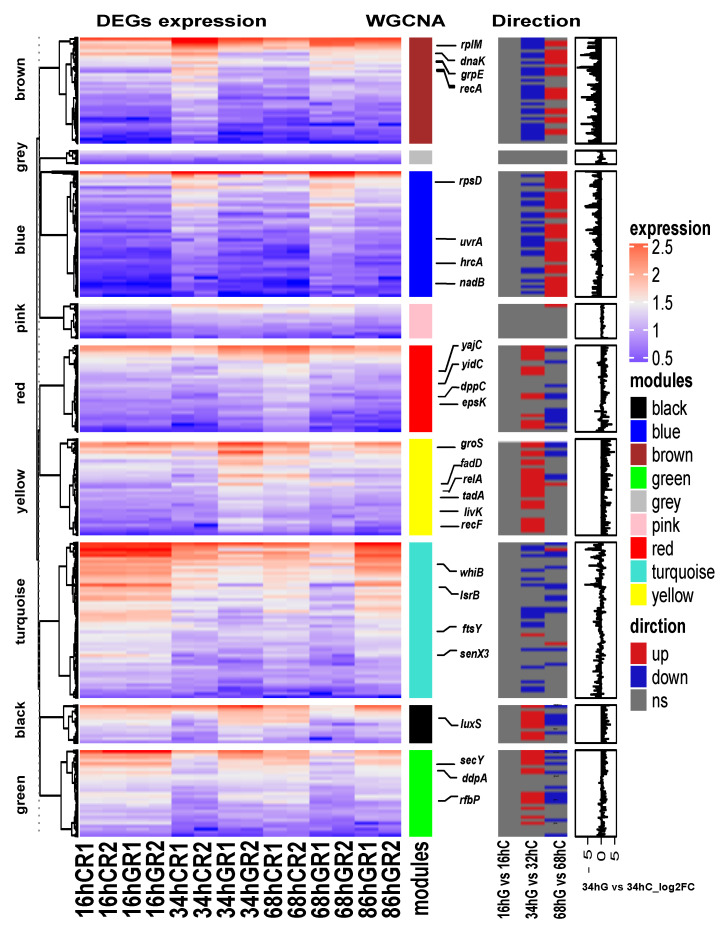
Heatmap and hierarchical clustering of the differentially expressed mRNAs. The left heat map shows the differentially expressed mRNAs in different samples. Red indicates a higher mRNA expression level, and blue indicates a lower mRNA expression level. The color from red to blue indicates log10 (fragments per kilobase of exon per million mapped fragments (FPKM) + 1) from high to low. The middle heat map shows the three direction of genes, upregulated, downregulated or no significance (ns). The right bar graph is the differential expression log2 fold change of genes in S2 stage (34 hG vs. 34 hC).

**Table 1 microorganisms-09-00385-t001:** Clusters and key DEGs genes related to the SOS response.

Cluster	Gene ID	Name	S1_FC ^1^	S2_FC	S3_FC	Function
4	Blo01|peg.321	*guaB*	−1.02 ^2^	2.08	−1.42	Plays an important role in the regulation of cell growth and oxidation-reduction process
	Blo01|peg.827	*ruvA*	−1.13	−3.25	−1.12	The RuvA-RuvB complex in the presence of ATP renatures cruciform structure in supercoiled DNA
	Blo01|peg.826	*ruvB*	−1.14	−2.66	1.44	
	Blo01|peg.851	*uvrA*	−1.02	−2.31	1.91	The UvrABC repair system catalyzes the recognition and processing of DNA lesions.
	Blo01|peg.97	*uvrB*	1.13	3.78	−1.34	
6	Blo01|peg.498	*groS*	1.04	3.91	−1.32	Regulation of transcription under stress conditions
	Blo01|peg.1014	*dnaK*	1.16	−2.10	3.15	Heat shock 70 kDa protein
	Blo01|peg.1272	*recA*	−1.12	−3.41	2.68	It interacts with LexA. Regulator of the SOS system
7	Blo01|peg.432	*radA*	−1.23	−2.57	1.02	Plays a role in repairing DNA breaks.
	Blo01|peg.1510	*recF*	−1.23	4.78	−1.46	Required for DNA replication and normal SOS inducibility
	Blo01|peg.47	*recN*	−1.09	−1.55	2.04	Involved in recombinational repair of damaged DNA
	Blo01|peg.1390	*recQ*	1.14	−1.78	2.58	ATP-dependent DNA helicase RecQ
	Blo01|peg.1273	*recX*	−1.25	−4.00	3.69	Modulates RecA activity
14	Blo01|peg.668	*pstA*	−1.00	1.70	−3.47	Phosphate transport system permease
	Blo01|peg.669	*pstB*	1.24	2.47	−1.60	Part of the ABC transporter complex PstSACB involved in phosphate import and Responsible for energy coupling
	Blo01|peg.666	*pstS*	1.22	1.96	−2.71	
15	Blo01|peg.223	*sufB*	−1.01	3.03	−3.27	FeS assembly protein SufB
	Blo01|peg.225	*sufC*	−1.05	3.42	−2.19	FeS assembly ATPase SufC
	Blo01|peg.224	*sufD*	−1.14	3.20	−2.46	Sulfur compound metabolic process and response to oxidative stress

^1^ FC: Fold Change. ^2^ Downregulated.

**Table 2 microorganisms-09-00385-t002:** The sequence identity of AI-2 receptor homologues in *B. longum* FGSZY16M3.

Gene ID	Protein	Acc. NO	Species	% identity	*E* Value
Blo01|peg.1718	rbsB	P36949.2	RBSB_BACSU	31.05	1.22 × 10^−30^
Blo01|peg.1719	rbsA	Q9KN37.1	RBSA_VIBCH	42.05	1.15 × 10^−123^
Blo01|peg.1720	rbsC	P44736.1	RBSC_HAEIN	33.44	2.16 × 10^−35^
Blo01|peg.1721	rbsC	P36948.2	RBSC_BACSU	35.61	9.72 × 10^−35^

**Table 3 microorganisms-09-00385-t003:** Expression of key biofilm regulatory genes.

Gene ID	Name	16 hG	34 hG	68 hG	86 hG
Blo01|peg.1391	*luxS*	73.04	120.65	32.13	110.59
Blo01|peg.1718	*rbsB*	387.78	287.46	178.82	13.67
Blo01|peg.1719	*rbsA*	27.75	15.94	7.67	5.14
Blo01|peg.1720	*rbsC*	20.30	16.25	5.48	3.62
Blo01|peg.1721	*rbsC*	30.93	25.59	8.18	7.00
Blo01|peg.664	*senX3*	5.97	5.01	3.749	7.37
Blo01|peg.665	*regX3*	9.48	9.58	11.18	9.78
Blo01|peg.1014	*dnaK*	503.24	617.10	482.19	495.50
Blo01|peg.498	*groS*	138.75	5237.28	524.69	602.13
Blo01|peg.478	*rplM*	920.08	410.42	461.66	816.59
Blo01|peg.76	*whiB*	1152.20	146.66	288.31	894.02
Blo01|peg.1751	*zur*	67.289	292.95	17.24	38.31

## Data Availability

All raw data for RNA-seq were deposited into NCBI (BioProject accession code PRJNA680454).

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
