# Peer review of "Transcriptome Analysis Reveals the Genes Involved in *Bifidobacterium Longum* FGSZY16M3 Biofilm Formation"

_microorganisms, 2021, doi:10.3390/microorganisms9020385_

Round 1
Reviewer 1 Report
The authors presented their nice work on studying the biofilm formation of bifidobacteria and identified several genes related to SOS response and EPS production. The conclusion is well supported by the experimental data and the manuscript is well prepared and easy to read.
Minor suggestions/comments
Line 50: please define SOS
Line 79: Please provide information on strain source, isolation, and deposition.
Line 91: what were the designations of 16h C, 16h G based on?
Line 148, biofilms on …; biofilms were not directly shown in Fig. 1, I would say cell aggregates were formed as cells were cultivated in liquid and GM were not clearly visible.
Line 168: 16h G….
Line 372: may be related to
Section 5. Conclusions: what could be the next steps to verify the roles of identified genes related to stress response and EPS production and a kind of outlook to this work might be suitable here.
Reviewer 2 Report
Please find my comments in the attached file.

Reviewer 3 Report
The manuscript is well-written, but I suggest to improve Introduction via extending this section.
Flasks on Figure 1 are not very demonstrative, I suggest to find photos that are more representative or to move them into the Supplementary.
The term "two-component systems" and its signaling role should be explained in the Introduction.
Line 91 - 16h C, 16h G, 34hC, 34h G, 68h C, 68h G, 86h G. It should be clarified that C - control, G - in the presence of glass microspheres (GM).
Table 1 - description of functions should be written in the same style and begin from the capital letter.
Lines 265 and 366. Please, check if ung gene is written correctly. In both lines "and" between typA and ung is written in italic.
Line 360 - "PPInetwork" should be written separately. Line 366 - ((Figure 5d)) - delete double parentheses.Author Response
Please see the attachment

Round 2
Reviewer 2 Report
Authors have addressed all my comments and suggestions . The present revised version could be considered for publication.